# Lidar-Derived Aerosol Properties from Ny-Ålesund, Svalbard during the MOSAiC Spring 2020

**Jonas Dube** [1,2], **Christine Böckmann** [1,3] **and Christoph Ritter** [1,*]

1   Alfred-Wegener-Institut, Helmholtz-Zentrum für Polar- und Meeresforschung, Telegrafenberg A45, 14473 Potsdam, Germany; jonas.dube@awi.de (J.D.); bockmann@uni-potsdam.de (C.B.)
2   Institut für Physik, Humboldt-Universität zu Berlin, Unter den Linden 6, 10099 Berlin, Germany
3   Institut für Mathematik, Universität Potsdam, Am Neuen Palais 10, 14469 Potsdam, Germany
*   Correspondence: christoph.ritter@awi.de; Tel.: +49-(331)288-2166

**Abstract:** In this work, we present Raman lidar data (from a Nd:YAG operating at 355 nm, 532 nm and 1064 nm) from the international research village Ny-Ålesund for the time period of January to April 2020 during the *Arctic haze* season of the MOSAiC winter. We present values of the aerosol backscatter, the lidar ratio and the backscatter Ångström exponent, though the latter depends on wavelength. The aerosol polarization was generally below 2%, indicating mostly spherical particles. We observed that events with high backscatter and high lidar ratio did not coincide. In fact, the highest lidar ratios (LR > 75 sr at 532 nm) were already found by January and may have been caused by hygroscopic growth, rather than by advection of more continental aerosol. Further, we performed an inversion of the lidar data to retrieve a refractive index and a size distribution of the aerosol. Our results suggest that in the free troposphere (above ≈2500 m) the aerosol size distribution is quite constant in time, with dominance of small particles with a modal radius well below 100 nm. On the contrary, below ≈2000 m in altitude, we frequently found gradients in aerosol backscatter and even size distribution, sometimes in accordance with gradients of wind speed, humidity or elevated temperature inversions, as if the aerosol was strongly modified by vertical displacement in what we call the "mechanical boundary layer". Finally, we present an indication that additional meteorological soundings during MOSAiC campaign did not necessarily improve the fidelity of air backtrajectories.

**Keywords:** aerosol; *Arctic haze*; lidar; microphysical properties; backtrajectories; Ny-Ålesund; Svalbard; MOSAiC; aerosol-boundary layer interactions

## 1. Introduction

The climate in the Arctic region is very vulnerable. The temperature increase in this region is faster than elsewhere on the globe—a phenomenon that is called *Arctic amplification* [1]. Understanding this increased Arctic warming is a relevant scientific question, as it may contribute to winter cooling in the mid-latitudes [2,3]. However, the role of aerosols in Arctic amplification is still not sufficiently known [4]. To fill the gap of observational data of the complex Arctic system, the international MOSAiC drifting experiment was performed between October 2019 and October 2020 [5]. During this campaign, numerous aerosol measurements were performed, e.g., by Raman lidar [6]. Ny-Ålesund is an international research village in Svalbard in the European Arctic at 78.9 North and 11.9 East, which has shown drastic winter warming of more than 2 °C per decade [7]. Long-term measurements of aerosol have been performed at this site for many years, both in situ [8,9] and by remote sensing techniques [10–12]. For this reason, the Ny-Ålesund data are very valuable to compare to MOSAiC in order to interpret the 2019/2020 conditions during drift in a broader context.

As an Arctic site, increased air pollution normally occurs in spring time, an effect which is called *Arctic haze* [13]. Early publications on the *Arctic haze* phenomenon have

pointed to an anthropogenic origin of aerosols due to industrial activities, caused by the increased lifetime of tropospheric aerosol in the dry and stably stratified Polar atmosphere of late winter/early spring [14,15]. However, it has been shown by aircraft measurements over Alaska that biomass burning also contributes to *Arctic haze* [16]. Hence, climate change may alter the aerosol composition in the Arctic via different sources and pollution pathways. While some information on the *Arctic haze* phenomenon from different ground-based sites has been published, e.g., [17], a full understanding of the vertical distribution and the optical properties of this aerosol is still missing.

In this work, we present Raman lidar data from Ny-Ålesund for the period January to April 2020. We will provide typical aerosol properties such as backscatter coefficient, lidar ratio and Ångstöm exponent in Section 3. Further, we show the results of the inversion of microphysical aerosol parameters from lidar data in Section 4. We observed quite variable size distributions of aerosol below ≈2500 m, while the atmosphere seemed to be cleaner and the remaining aerosol much more constant above that altitude. First of all, we will give a short introduction in the next section of the instruments and methods used .

## 2. Instruments, Methods and Data

The remote-sensing data in this paper were measured with the Koldeway Aerosol Raman Lidar (KARL) at the French–German AWIPEV research station in Ny-Ålesund, Svalbard. A Spectra 290/50 Nd:YAG laser emits at 355 nm, 532 nm and 1064 nm, with 50 Hz and 200 mJ per pulse and color, vertically into the atmosphere. The backscattered light is collected by a 70 cm receiving telescope (with about 2 mrad field of view). Apart from the elastic backscattered photons, the vibrational Raman lines at 387 nm and 607 nm from the nitrogen molecules for the calculation of the extinction, and at 407 nm from $H_2O$ molecules) for the determination of the absolute humidity are also measured. Further, the system consists of Licel transient recorders with 7.5 m resolution and 12-Bit A/D conversion for the elastic and 16-Bit for the inelastic scattered signals. For the detection of the photons, Hamamatsu photomultipliers are used. An APD from Licel is employed only for the 1064 nm channel. As we reach complete overlap of the laser beam and the telescope at 700 m, we will only discuss data above that altitude. A more detailed description of the instrument is given in [18]. The calculations of the aerosol backscatter coefficient $\beta^{aer}$ (unit: $Mm^{-1}sr^{-1}$) and the aerosol extinction coefficient $\alpha^{aer}$ (unit: $Mm^{-1}$) were performed as described by [19]. The channel at 1064 nm was evaluated according to [20] with a Lidar ratio of 45 sr, and LR = 30 sr for those times and altitudes at which clouds were detected. The extinction was determined for 355 nm and 532 nm, while the backscatter coefficients were established at all elastic wavelengths. The lidar ratio *LR* (unit: sr), aerosol depolarization $\delta^{aer}$ and the backscatter Ångström exponent (Å) (In the following, "Ångström exponent" for brevity) were also calculated as described in Equation (1). Here, $\beta^{aer}_{\perp}$ and $\beta^{aer}_{\parallel}$ are the backscatter coefficients with respect to the perpendicular and parallel laser polarization, respectively, which are measured separately in the elastic 355 nm and 532 nm channels. In this paper, for $\beta^{aer}$ and $\delta^{aer}$, the 532 nm signal and for *LR* the 355 nm channel were used, unless otherwise stated. For the calculation of the Ångström exponent, we used the 355 nm and 532 nm lines.

$$LR(\lambda) = \frac{\alpha^{aer}(\lambda)}{\beta^{aer}(\lambda)} \qquad \delta^{aer}(\lambda) = \frac{\beta^{aer}_{\perp}(\lambda)}{\beta^{aer}_{\parallel}(\lambda)} \qquad \mathring{A} = \frac{\log\left(\frac{\beta^{aer}(\lambda_1)}{\beta^{aer}(\lambda_2)}\right)}{\log\left(\frac{\lambda_2}{\lambda_1}\right)} \qquad (1)$$

Classification of different aerosol types is usually done by using the lidar ratio. Typically, values between $LR = (20 - 35)$ sr indicate maritime aerosol or clouds, and larger values around $LR = (45 - 70)$ sr refer to continental aerosol [21]. The depolarization allows a rough assessment of the particle shape, since aspherical particles generally change the state of polarization of backscattered photons. With the used lidar, the Rayleigh scattering of clear air has a depolarization of 1.43% [22], so we can also expect spherical particles at values in this range. The Ångström exponent (or similarly, the color ratio) is often used

for an estimation of the particle size. On the basis of the Mie theory, we expect a high wavelength dependence of the backscatter coefficient $\beta^{\text{aer}}$ for small particles ($\mathring{A} \approx 4$) and no dependence for large particles ($\mathring{A} \approx 0$) in relation to the wavelength used. The data were evaluated with a resolution of 11.5 min and 7.5 m. Overall, 2954 time steps between 9 January and 26 April were analyzed, which roughly correspond to 1/5 of data coverage. Due to high cloud cover in the European Arctic and safety regulations of the local air field, this is already very good data availability for our station. Further, we applied a cloud mask in such a way that all data with a backscatter ratio $BSR > 3$ were deleted, to avoid bias in the aerosol statistics. At the value of $BSR > 3$, a strong increase of the particle size (indicated by the Ångström exponent) was observed, and the value coincides with earlier studies in Ny-Ålesund., e.g., [12]. At the same time, we ruled out effects of multiple scattering. Hence, the highest remaining backscatter ratio $BSR$ at 532 nm was $BSR = 3$. Due to polar stratospheric clouds, the boundary condition was adjusted for some days to fulfill the clear sky approximation.

Information on the relative humidity $RH$ and temperature $T$ was taken from Vaisala RS-41 radiosondes, which are launched at least once per day from Ny-Ålesund. Due to the MOSAiC expedition, there were up to four launches per day in February and March. When lidar and radiosonde data were compared, only data within $\pm 30$ min to the radiosonde launch were considered, so we can expect no large changes in the atmosphere during that time.

### 2.1. Microphysical Retrieval Methodology by Regularization

In this subsection, the retrieval of microphysical aerosol properties from lidar data is described. The model relating the optical parameters $\Gamma(\lambda)$ with the particle volume size distribution (PVSD) $v(r)$ is described by a Fredholm integral operator of the first kind:

$$\Gamma(\lambda) = \int_{r_{\min}}^{r_{\max}} K(r, \lambda; m) v(r) \, \mathrm{d}r \tag{2}$$

with the kernel function $K(r, \lambda; m) = \frac{3}{4r} Q(r, \lambda; m)$, where $\lambda$ is the wavelength, $r$ is the radius, $r_{\min}$, $r_{\max}$ are suitable lower and upper radius limits, $m$ is the complex refractive index (CRI), $\Gamma(\lambda)$ denotes either the mean extinction or backscatter coefficients of the regarded layer and $Q$ stands for either the extinction or the backscatter (dimensionless) Mie efficiencies. Since from a mathematical point of view this operator, which is mapping here between Hilbert spaces, is compact, we have to deal with an inverse ill-posed problem. Identifying $\Gamma(\lambda)$ as our input data and $v(r)$ as the unknown PVSD, the problem reduces to the inversion of Equation (2). Knowing the PVSD, we can extract the following microphysical parameters:

- Surface-area concentration (first ("fine") mode, second ("coarse") mode, total) ($\mu m^2 cm^{-3}$)
  $s_{\text{t}} = 3 \int \frac{v(r)}{r} \mathrm{d}r$;
- Volume concentration (first mode, second mode, total) ($\mu m^3 cm^{-3}$)
  $v_{\text{t}} = \int v(r) \mathrm{d}r$;
- Number concentration (first mode, second mode) ($cm^{-3}$)
  $n_{\text{t}} = 3/4\pi \int \frac{v(r)}{r^3} \mathrm{d}r$;
- Effective radius ($\mu m$)     $r_{\text{eff}} = 3 \frac{v_t}{s_t}$.

In addition, a mean complex index of refraction (CRI) and the single scattering albedo (SSA) at 355 nm and 532 nm are retrieved. A wavelength-independent CRI is assumed as a member of a predefined grid; see Figure 1. Solving Equation (2) requires discretization, regularization and a parameter choice rule; see e.g., [23–26].

We discretize Equation (2) with spline collocation. The PVSD $v(r)$ is approximated with respect to the B-spline functions $\phi_j$ by $v_n = \sum_{j=1}^{n} b_j \phi_j$, reducing the problem to the determination of the coefficients $b_j$. The infinite-dimensional ill-posed problem (Equation (2)) is thereby replaced by a finite-dimensional quite often ill-conditioned one, namely $Ab = \Gamma$. The matrix elements of the linear system $Ab = \Gamma$ are determined by

$$A_{ij} = \int_{r_{\min}}^{r_{\max}} K(\lambda_i, r; m) \phi_j(r) \mathrm{d}r \qquad (3)$$

and $\Gamma$ is now a vector. By doing this, we have projected our problem to a finite n-dimensional space. Clearly, the decision about the projected dimension ($n$) and order ($d$) of the base functions $\phi_j$ is critical, since they act as regularization parameters, and therefore an appropriate selection is necessary.

The software for the retrieval of microphysical aerosol parameters from lidar data, (see [26]) provides two different regularization techniques. The first one is Truncated Singular Value Decomposition (TSVD) [23], whereas the second one is an iterative regularization; here, a generalized Runge–Kutta iteration [24], also known as a Padé (PADE) iteration [27], is provided.

Using the TSVD technique, the linear system is solved by first expanding the matrix using SVD. Potential noise in our matrix will be magnified as a result of the ill-conditionedness of the matrix $A$, i.e., the singular values clustering to zero. We would like to prevent this behavior by including only a part of the SVD, i.e., defining a certain cut-off level $k$, below which the noisiest solution coefficients are filtered out. Mathematical parameter choice rules for $k$ may include a known noise level. Unfortunately, those rules are often very sensitive with respect to under- or overestimated noise levels. Since, in almost all cases, the noise level of the backscatter and extinction values is only known approximately, the appropriate triple $(n, d, k)$ is selected from simulation knowledge [26]. Recently, in [28], an interesting new possibility using a Bayesian model and a Monte Carlo algorithm was proposed and tested on synthetically generated data.

Using the PADE technique, the linear system is solved by an iteration with the initial value $b_i = 0, i = 0$. While for well-conditioned coefficient matrices the iteration may proceed "nearly" to infinity, in our case, one must terminate the iteration early enough to suppress the amplification of the noise. One must now use an appropriate triple $(n, d, i)$. Additionally, here the base points on the radius axis are increasing during the iteration and are shifting automatically from an initial equidistant to a non-equidistant grid to be able to model strong slopes of the size distribution function. Moreover, the iteration is a projected iteration, meaning that potentially negative values of the PVSD during the iteration are projected to zero. For the case studies in Section 4, we deal with PADE ($i = 100$) iteration retrieval.

Since the CRI is also an unknown variable, the software deals with a RI grid of possible real and imaginary parts of the CRI; see Figure 1. The potential candidates of appropriate CRI are quite often located in a diagonal pattern, which is an indication of good optical data quality. Moreover, this means that the CRI is not uniquely for retrieval. We only obtain a mean value of the CRI. Figure 1 shows an example of which points on the diagonal were selected. Roughly speaking, a cluster of physically meaningful grid points, which delivers a similar shape of the PVSD, was selected. It is worth mentioning that the shape retrieval of the PVSD was without any restrictions to the number of modes and the shape of the curve, as is done in [28] with a known refractive index as input.

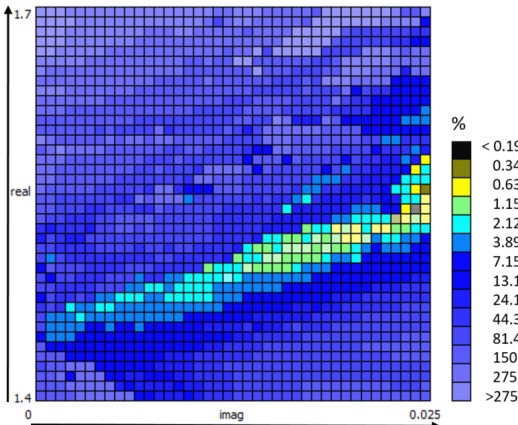

**Figure 1.** Grid of the imaginary and real part of the CRI at 21 February 2020, lower layer from 1323 until 1586 m. The selected grid points in the diagonal pattern to determine the mean CRI are marked brightly.

## 3. Aerosol Properties in Spring 2020

To present an overview, frequency polygons of the measured backscatter coefficients $\beta^{aer}$ at 532 nm (for brevity) for different months and heights are plotted in Figure 2. The altitude ranges were chosen in such a way that they represent the atmosphere as best as is possible, meaning that there are only small changes within each range. The histogram classes become wider for larger values and are represented by the grid. For better comparison, previously published values for January–March 2019 are also shown in grey [29]. In the lowest troposphere up to 1500 m, there were much higher backscatter values in 2020 than in 2019. The mechanical boundary layer (MBL) is usually located at this altitude, which is described in more detail in Section 4.2. In the undisturbed troposphere, the values are very similar in both years, with slightly increased values in 2020 in the highest troposphere above 8 km. This is caused by a higher aerosol concentration in the stratosphere due to wildfire smoke [6].

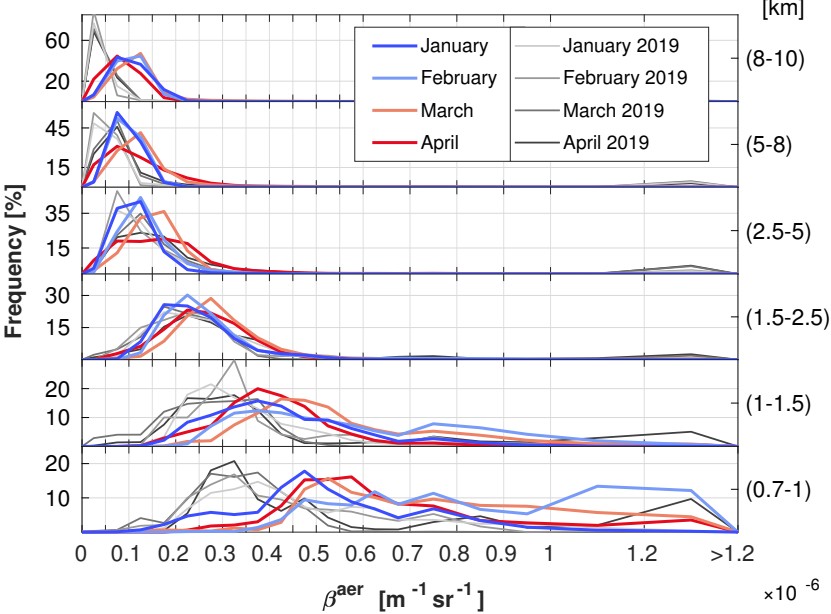

**Figure 2.** Frequency polygon of the backscatter coefficient $\beta^{aer}$ (532 nm) for different altitude ranges (labels on the right side) and months (indicated by colors). As a reference, the data from [29] for 2019 are plotted in grey.

In Figure 3, the temporal evolution of the backscatter coefficients is shown for the different altitude ranges. The solid line represents the median value and the error bar is the 25th and 75th percentiles as an indicator of the distribution width. The median was used since it is more stable regarding outliers and events with strong increased backscatter. Therefore, it gives a better overview of the measured values than the mean does. As expected, the backscatter decreases with height, and higher values also show increased variability. As the backscatter coefficients can physically only be positive, the 75th percentile always differs more from the median than the 25th percentile. It can be seen that the maximum of $\beta^{\text{aer}}$, which we call *Arctic haze*, occurs in February at (0.7–1) km, while this shifts to March or even April in the upper troposphere. Usually, it is expected that *Arctic haze* is mainly caused by a higher amount of continental aerosol, meaning a high lidar ratio [30]. Due to the strong noise when evaluating the extinction in lidar, we only considered the LR up to 1.5 km in the same altitude ranges as before. The median values and percentiles are presented in Table 1 separately for all high backscatter events ($\beta^{\text{aer}}$ larger than median) and all low backscatter events ($\beta^{\text{aer}}$ smaller than median).

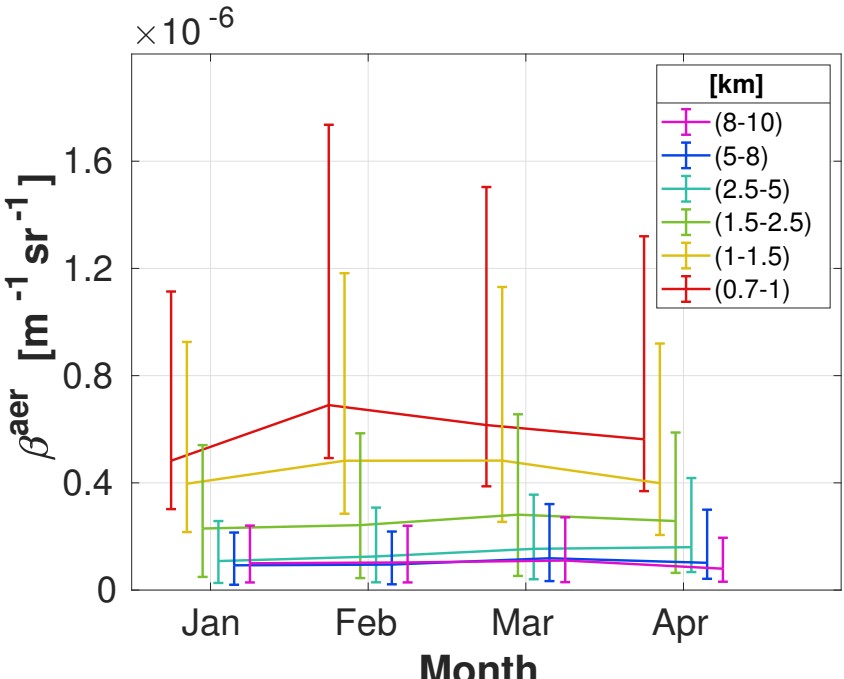

**Figure 3.** Backscatter coefficient $\beta^{\text{aer}}$ (532 nm) over time for different altitude ranges. The median value is plotted (different colors mark altitude ranges). The error bar is the mean value of all values larger and smaller than the median.

**Table 1.** Median of the lidar ratio *LR* for different altitude ranges and months in 2020, with high and low backscatter events presented separately. The 25th and 75th percentiles are indicated in the brackets.

| High | January | February | March | April |
|---|---|---|---|---|
| (0.7–1) km | 30.2 (20.1–49.8) | 27.1 (20.2–34.7) | 22.1 (12.6–33.7) | 30.9 (20.6–40.3) |
| (1–1.5) km | 32.3 (16.3–52.6) | 24.9 (11.6–39.5) | 21.9 (7.0–42.2) | 26.5 (15.0–38.6) |
| **Low** | **January** | **February** | **March** | **April** |
| (0.7–1) km | 54.3 (35.7–79.4) | 33.7 (20.0–47.2) | 27.8 (11.4–53.0) | 28.6 (15.0–40.7) |
| (1–1.5) km | 58.5 (32.6–82.0) | 29.1 (7.3–54.0) | 16.3 (0.9–38.9) | 29.1 (10.3–50.3) |

An increased lidar ratio occurred in January, with the largest values during low backscatter events, as can be seen in Table 1. An increased lidar ratio during high backscatter

events could not be observed. Hence, there is not one clear *Arctic haze* event, and the higher amount of continental aerosol and the increased backscatter coefficients may have to be distinguished as two different occurrences. Therefore, for this paper, we define the *Arctic haze* simply as an increased backscatter event independent of the microphysical properties of the aerosol. During the whole period, the mean value of the depolarization, calculated per month and height interval, was always below 2%, except for in March and April between (2.5–8) km, where values up to 2.5% were measured (not shown). Furthermore, with these values we expect predominantly spherical particles. The Ångström exponent *Å* was calculated from the elastic backscatter signals as is described in Equation (1). Closer inspection showed that it should be calculated by two wavelengths that are not too far away, which will be described in more detail in Appendix A, in our case 355 nm and 532 nm.

The strongest *Arctic haze* events (in terms of high backscatter coefficients) were observed in February at (0.7–1) km, with a bimodal occurrence distribution of the measured backscatter coefficients as can be seen in Figure 2. The high backscatter mode mainly occurred between 21 to 23 February, with the highest values on 21 February. As can be seen in Table 2, the backscatter values of these three days differ strongly from the rest of February, while the other optical parameters are quite similar. To make the comparison more stable to single outliers (especially for the LR), the median was used with the 25th and 75th percentiles, indicating the spread of the measured values. In both modes, the depolarization is smaller than $\delta^{aer} < 2\%$, so we expect spherical particles. The median values of the lidar ratio *LR* are also very similar. The percentiles indicate a larger spread of the *LR* for the low backscatter mode due to a larger data set, while the high backscatter mode contains only three days. Since the *LR* and $\delta^{aer}$ of both modes indicate similar aerosol properties, we expect that the differences in the backscattering are mainly attributable to different surface and number concentrations.

**Table 2.** Median of the retrieved optical parameters shown separately for the 3 days in February 2020 with the highest backscatter and for the rest of February. The 25th and 75th percentiles are indicated in the brackets.

| Date | High Backscatter Mode 21–23 February | Low Backscatter Mode Rest of February |
|---|---|---|
| $\beta^{aer}_{532}$ | 1.12 (0.98–1.23) | 0.69 (0.56–1.01) |
| $LR_{355}$ | 28.9 (23.6–36.7) | 29.5 (18.3–41.9) |
| $\delta^{aer}_{532}$ [%] | 0.65 (0.57–0.74) | 0.85 (0.74–1.02) |
| $Å(355, 532\,\text{nm})$ | 1.04 (0.83–1.22) | 0.96 (0.56–1.18) |

## 4. Case Studies

As already mentioned above, events with a high backscatter coefficient and with a high lidar ratio must be distinguished. To obtain more information about the similarities and differences between the aerosols in the events with a high backscatter coefficient and with a high lidar ratio, two representative cases are presented in detail. As an example of a high backscatter event, we take 21 February 2020. On this day, we measured the highest backscatter coefficients at an altitude of (700–1000) m. To obtain stable results for the calculation of the microphysical parameters, we have chosen a slightly higher altitude for our further studies. 13 January is an example of a day with a high lidar ratio. With HYSPLIT, we calculated 5-day backward trajectories with the model *GDAS1*, which can be seen in Figure 4. The chosen start position is the location of the lidar in Ny-Ålesund at an altitude we investigated in more detail. The trajectories were calculated with the ensemble method, so the start points vary on a three-dimensional grid. Due to the small amount of meteorological data in the Arctic region, the trajectories should be used with caution, as is described in more detail in Appendix B. It is shown in Appendix B that the trajectory on 13 January had a larger uncertainty, but we expected air masses from the Eurasian continent on both days. The air masses on 13 January crossed the position of the research vessel *Polarstern* during the MOSAiC expedition about 29 h before. The position

of the vessel at this time is also shown in Figure 4. Due to the different behavior of the optical parameters, we expect distinct aerosol conditions at both events. Hence, it is clear that the origin of the air masses is not sufficient to explain these differences. To obtain more information, we calculated the microphysical properties for two different layers on both days via the inversion method described in Section 2.1. One layer is located within the "mechanical boundary layer" (see Section 4.2) and the other one above, in the free troposphere. Because of differences in the height of the MBL, the chosen altitudes vary for both days. In the next subsection, we will discuss the differences between the two days, and in Section 4.2, we will describe in more detail the effect of the MBL. All optical and microphysical parameters of the layers are collected in Table 3. The optical parameters in this table are the mean value of the specified time period and height interval. The uncertainties of the optical coefficients have been calculated according to the maximum error estimate, i.e., the error of backscatter or extinction in the sum of the absolute amount of the individual error sources.

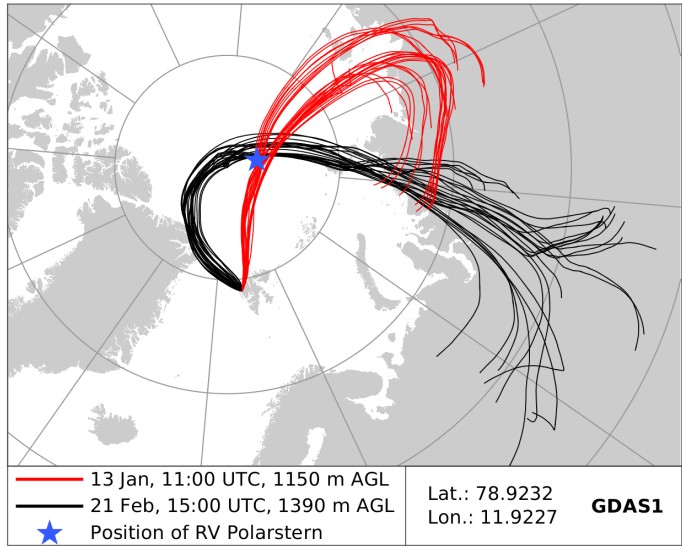

**Figure 4.** Five-day backward trajectories calculated with HYSPLIT [31,32]. The initial position is the location of the lidar in Ny-Ålesund at the altitude of the sampled aerosol layer. The position of RV Polarstern at 12 January, 2020, 06:00 UTC is also marked [33].

**Table 3.** Optical and microphysical parameters for the different layers. The first mode is always the mode with the smaller radius $r_{\mathrm{mod}}$. For the optical parameters, the largest possible uncertainty according to the maximum error estimate is given.

| | **13 January** | | **21 February** | |
|---|---|---|---|---|
| Time [UTC] | 10:21–12:14 | 10:21–12:14 | 13:40–15:10 | 13:40–15:10 |
| Height [m] | 1050–1250 | 1600–1900 | 1323–1586 | 2150–2750 |
| $\beta_{355\,\mathrm{nm}}^{\mathrm{aer}}$ [Mm$^{-1}$sr$^{-1}$] | 1.044 ± 0.08 | 0.465 ± 0.08 | 1.077 ±0.08 | 0.277 ± 0.07 |
| $\beta_{532\,\mathrm{nm}}^{\mathrm{aer}}$ [Mm$^{-1}$sr$^{-1}$] | 0.545 ± 0.05 | 0.277 ± 0.05 | 0.7414 ± 0.04 | 0.207 ± 0.02 |
| $\beta_{1064\,\mathrm{nm}}^{\mathrm{aer}}$ [Mm$^{-1}$sr$^{-1}$] | 0.244 ± 0.02 | 0.092 ± 0.02 | 0.219 ± 0.02 | 0.029 ± 0.02 |
| $\alpha_{355\,\mathrm{nm}}^{\mathrm{aer}}$ [Mm$^{-1}$] | 35.912 ± 8 | 20.777 ± 9 | 40.848 ± 8 | 12.650 ± 7 |
| $\alpha_{532\,\mathrm{nm}}^{\mathrm{aer}}$ [Mm$^{-1}$] | 42.287 ± 19 | 5.343 ± 2.1 | 29.144 ±16 | 5.331 ± 2.3 |
| $LR_{355\,\mathrm{nm}}^{\mathrm{aer}}$ [sr] | 34.40 (24.9–45.8) | 44.68 (23.3–81.2) | 37.93 (28.4–49.0) | 44.28 (16.3–94.9) |
| $LR_{532\,\mathrm{nm}}^{\mathrm{aer}}$ [sr] | 77.59 (38.4–122.4) | 19.29 (9.6–38.8) | 39.33 (16.8–64.4) | 25.79 (13.3–40.8) |
| Mean RI$_{\mathrm{real}}$ | 1.311 ± 0.010 | 1.458 ± 0.009 | 1.526 ± 0.015 | 1.447 ± 0.011 |
| Mean RI$_{\mathrm{imag}}$ | 0.0006 ± 0.0005 | 0.0010 ± 0.0004 | 0.020 ± 0.004 | 0.0039 ± 0.0017 |
| Total: $r_{\mathrm{eff}}$ [μm] | 0.73 ± 0.06 | 0.053 ± 0.003 | 0.0799 ± 0.0033 | 0.0538 ± 0.0014 |

**Table 3.** *Cont.*

|  | 13 January | | 21 February | |
|---|---|---|---|---|
| First mode: $r_{\text{mod}}$ [μm] | 0.54 | 0.016 | 0.004 | 0.015 |
| Second mode: $r_{\text{mod}}$ [μm] | 1.44 | 2.26 | 0.53 | 0.69 |
| First mode: $\sigma$ | 1.38 | 2.03 | 3.19 | 2.16 |
| Second mode: $\sigma$ | 1.16 | 1.09 | 1.14 | 1.09 |
| First mode: $v_{\text{t}}$ [μm$^3$cm$^{-3}$] | 11.06 | 4.80 | 5.04 | 1.98 |
| Second mode: $v_{\text{t}}$ [μm$^3$cm$^{-3}$] | 2.68 | 2.17 | 1.17 | 0.65 |
| Total: $v_{\text{t}}$ [μm$^3$cm$^{-3}$] | 14.01 ± 1.22 | 7.27 ± 0.28 | 6.16 ± 0.19 | 2.90 ± 0.13 |
| First mode: $s_{\text{t}}$ [μm$^2$cm$^{-3}$] | 47.36 | 261.14 | 143.52 | 90.15 |
| Second mode: $s_{\text{t}}$ [μm$^2$cm$^{-3}$] | 5.32 | 2.83 | 6.37 | 2.79 |
| Total: $s_{\text{t}}$ [μm$^2$cm$^{-3}$] | 57.24 ± 0.86 | 409.2 ± 14.53 | 231.66 ± 9.56 | 161.55 ± 0.13 |
| First mode: $n_{\text{t}}$ [cm$^{-3}$] | 10.46 | 30 899 | 58 420 | 9 848 |
| Second mode: $n_{\text{t}}$ [cm$^{-3}$] | 0.20 | 0.04 | 1.76 | 0.47 |
| SSA$_{355}$ | 0.9863 ± 0.0013 | 0.98637 ± 0.00025 | 0.854 ± 0.012 | 0.960 ± 0.004 |
| SSA$_{532}$ | 0.9917 ± 0.0006 | 0.9777 ± 0.0007 | 0.862 ± 0.012 | 0.946 ± 0.006 |

*4.1. High Backscatter vs. High Lidar Ratio*

For an overview of the lidar parameters, we show the measured optical parameters as altitude profiles in Figure 5a,b. The aerosol backscatter coefficients, extinction coefficients, the depolarization, lidar ratio and Ångström exponent are shown. The data is shown for 13 January (10:21–12:14) UT and 21 February (13:40–15:10) UT. In the figures, the time averaged profiles are presented. The altitude intervals for which the microphysical aerosol properties have been derived are marked with black lines.

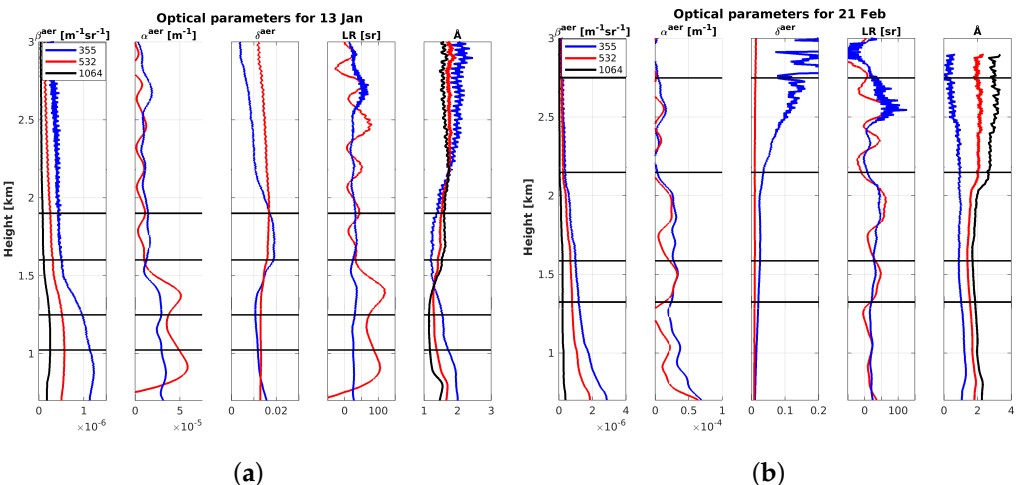

(**a**)  (**b**)

**Figure 5.** Profiles of the optical parameters for both case studies. The backscatter coefficient $\beta^{aer}$, aerosol extinction $\alpha^{aer}$, aerosol depolarization $\delta^{aer}$, lidar ratio LR and Ångström exponents Å are shown. The Ångström exponent is shown for all wavelength pairs: Å(355, 532 ) in blue, Å(355, 1064) in red, Å(532, 1064) in black. The investigated altitude intervals are marked with black lines. (**a**) Parameters from 13 January (10:21–12:14) UT. (**b**) Parameters from 21 February (13:40–15:10) UT.

Since the intensive optical parameters such as the depolarization and the lidar ratio were constant over the whole period, we expect similar aerosol properties for the entire time in the free troposphere (higher layer), including during *Arctic haze* events. This can also be seen by the microphysical parameters of both case studies, shown in Table 3. Since it is hardly possible to include the uncertainties of the optical measurements in the retrieval software, the uncertainties of the microphysical parameters only contain the uncertainty

from the inversion method and are therefore underestimated. In Appendix C, we give an example of how the errors from the measured optical parameters influence the result of the inversion. In Table 3, for the higher aerosol layers, the real parts of the refractive indices are equal within their given uncertainties with a value of $\text{RI}_{\text{real}} \approx 1.45$; the imaginary part is also in the same order of magnitude and ranges between $\text{RI}_{\text{imag}} = (0.0010\text{–}0.0039)$. As can be seen in Figures 6 and 7, there is a predominant first mode of small particles on both days with almost the same radius $r_{\text{mod}} \approx 0.016\,\mu\text{m}$ and $\sigma \approx 2.1$. On both days, this first mode differs in the concentration number, and therefore also in the volume and surface density. The second mode of larger particles is more distinct, with a radius of $r_{\text{mod}} = 2.26\,\mu\text{m}$ on 13 January and $0.69\,\mu\text{m}$ on 21 February, but these differences have only a small impact on the optical parameters, since in Figures 6 and 7, the volume distribution, not the size distribution, is depicted. Overall, the effective radii of both aerosol distributions are equal within their given uncertainty, with a value of $r_{\text{eff}} \approx 0.053\,\mu\text{m}$. It becomes apparent that in the free troposphere, there is a very stable mode of small particles with a small amount of more distinct larger particles. The uniform optical parameters indicate that this assumption is valid not only for both case studies, but for the whole period.

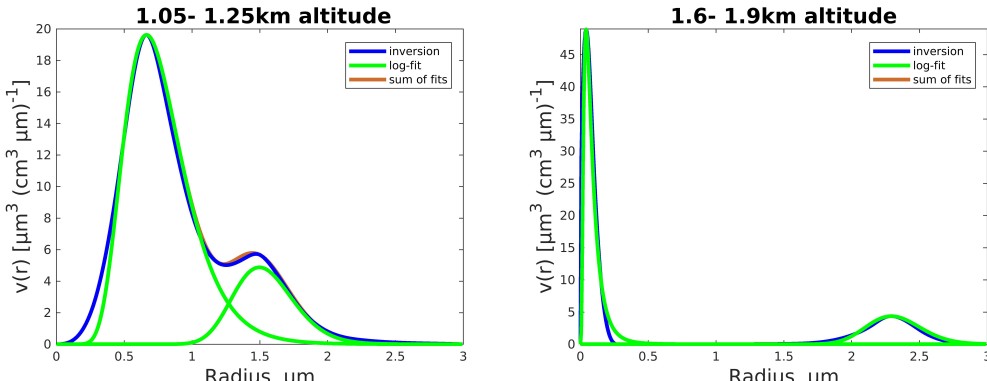

**Figure 6.** Volume distribution function of the particle radius $r$ for 13 January for two different altitudes, one within the MBL and one above. Blue: result of the inversion; green: log-normal fits. If the sum of the fits (in red) cannot be seen, it is congruent with the inversion. All values are given in Table 3. The mode with the smaller radius $r_{\text{mod}}$ is referred to as the *first mode*.

While there are constant values of the aerosol in the free troposphere, there are huge differences in the lower atmosphere. These differences are already indicated by the optical parameters. The lidar ratio on 13 January of $LR^{\text{aer}}_{532\,\text{nm}} = 77.59\,\text{sr}$ may suggest a high amount of continental aerosol, whereas we have a much smaller lidar ratio of $LR^{\text{aer}}_{532\,\text{nm}} = 39.33\,\text{sr}$ on 21 February. Hence, the question arises of to what extend different aerosol types were present at both events. A first difference is found in the refractive index. On 13 January, we have a smaller value of $\text{RI}_{\text{real}} = 1.311 \pm 0.010$ than in the free troposphere. On 21 February, the value of $\text{RI}_{\text{real}} = 1.526 \pm 0.015$ is slightly higher than that of the layer above. On 13 January, the imaginary part of the refractive index is $\text{RI}_{\text{imag}} = 0.0006 \pm 0.0005$ and on 21 February it is $\text{RI}_{imag} = 0.020 \pm 0.004$. Another difference can also be seen in the distribution functions shown in Figures 6 and 7. On 13 January, we have two modes with a radius of $r_{\text{mod}} = 0.54\,\mu\text{m}$ and $r_{\text{mod}} = 1.44\,\mu\text{m}$, while on 21 February, we have two modes with $r_{\text{mod}} = 0.004\,\mu\text{m}$ and $r_{\text{mod}} = 0.53\,\mu\text{m}$. Hence, there is a much larger effective radius of $r_{\text{eff}} = 0.73 \pm 0.06\,\mu\text{m}$ on 13 January, while on 21 February, the value is much smaller, with $r_{\text{eff}} = 0.0799 \pm 0.0033\,\mu\text{m}$. The large density number with smaller particles on 21 February leads to a much higher surface concentration, but a smaller volume concentration. The low "water-like" refractive index, together with the larger effective radius on 13 January, raises the question of whether hygroscopic growth of the aerosol has possibly been observed. We will discuss this in the next section. In total, there is a strong variability of the microphysical and optical properties of the aerosol within the mechanical boundary layer.

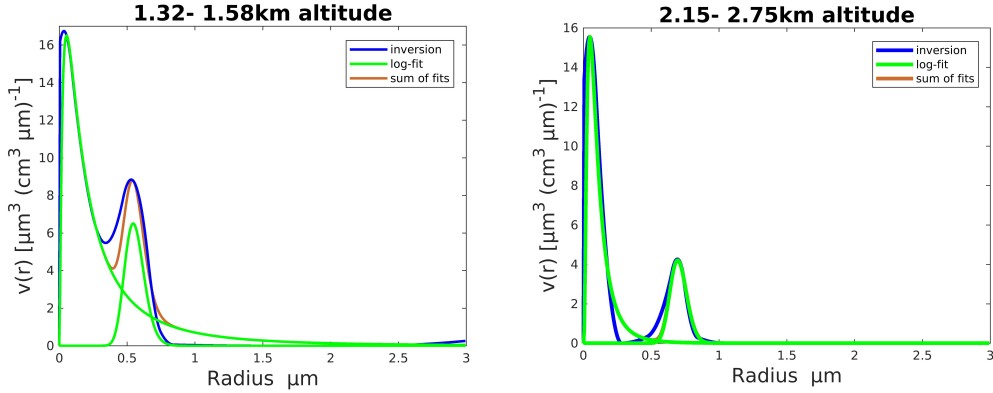

**Figure 7.** Volume distribution function of the particle radius *r* for two different altitudes on 21 February, one within the MBL and one above. Blue: result of the inversion; green: log-normal fits. If the sum of the fits (in red) cannot be seen, it is congruent with the inversion. All values are given in Table 3. The mode with the smaller radius $r_{\text{mod}}$ is referred to as the *first mode*.

### 4.2. Aerosol Properties in the Mechanical Boundary Layer

As is already shown in Section 4.1, the optical and microphysical properties of the aerosol are very distinct at various altitudes. The biggest changes typically arise between (1500–2500) m and are caused by what we call the *mechanical boundary layer* (MBL). Due to the cold ground, missing sunlight and, therefore, a thermally generally stable atmosphere, the regular planetary boundary layer usually ends just a few hundred meters above the ground in the Arctic during winter. However, in Svalbard, a mechanical mixing of the air masses also occurs, caused by wind shear due to mountains [34]. This is described by the concept of the bulk Richardson number [35]. Within this mechanical boundary layer, the air can be vertically displaced and come in contact with the ground. Hence, deposition of aerosols can occur and local aerosols could also be found. Nevertheless, we expect a predominant quantity of long-range transported aerosol even within the MBL during our case studies, due to a lack of local aerosol sources in winter [9]. During the launch time of the radiosonde, the MBL ended at ≈1500 m above ground on 13 January and at ≈2100 m on 21 February, as can easily be seen in Figures 8 and 9. The boundary layer height was determined using the backscatter coefficient. Below the edge of the MBL, a pronounced gradient of the backscatter coefficient is typically visible, which is lower and more constant over time in the free troposphere. This change is often accompanied by a shear of the wind direction (not shown) and an elevated temperature inversion. Due to the vertical mixing, there is also the most moisture in this layer, and therefore a strong gradient of relative humidity *RH* at the edge of the MBL. This can increase the backscatter and extinction coefficient in the area, caused by hygroscopic growth of the aerosol particles. All these effects, namely larger particle size, hygroscopic growth and overall different microphysical properties, lead to an increase of the backscatter and the extinction below ≈2000 m. Due to all these different effects, it is hard to separate them to obtain information about the contribution of a single factor. Furthermore, the assumption of a homogeneous mixing of the air masses is usually not valid within the mechanical boundary layer, as the wind shear pattern may be variable in time [34]. We speculate that this interaction of the aerosol with local gradients of wind, moisture and temperature in the lowest ≈2000 m is one reason for the typically complex vertical gradients of aerosol observed over Svalbard [36].

Our two case studies (at low altitude, within the MBL) took place at relatively moist conditions: at *RH* ≈ 65% on 13 January (large particles, low CRI) and at *RH* ≈ 55% on 21 February. Based on in situ measurements, it was found that in Ny-Ålesund, relative humidities around 50% are sufficient for hygroscopic growth of aerosol [37]. Even if a final proof is still missing, the low refractive index, the large effective radius and the high relative humidity for 13 January suggests that hygroscopic growth had occurred. Therefore, we assume that humidity in the MBL is one important factor that explains the variability

concerning lidar ratio and other optical parameters. Given the similar air backtrajectories for the two cases, different lidar ratios must not only be caused by advection of various aerosol types, but simply by water uptake of the same aerosol.

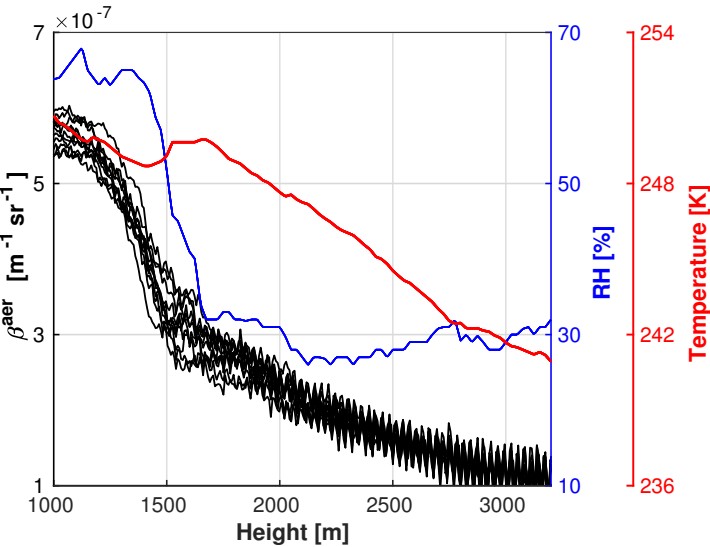

**Figure 8.** Atmospheric conditions on 13 January 2020. Meteorological data RH (in blue) and temperature (in red) from RS launch at 11:00 UTC, and lidar data $\beta^{aer}$ from (10:21–12:14) UTC (in black). The resolution of the lidar data is 10 min and 7.5 m.

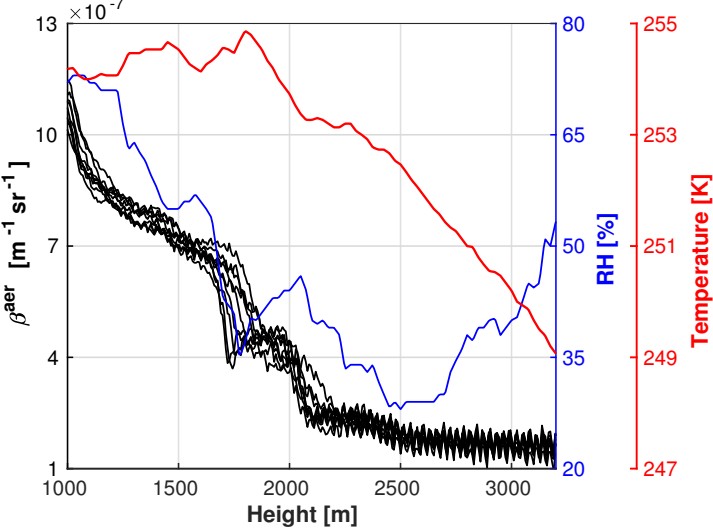

**Figure 9.** Atmospheric conditions on 21 February 2020. Meteorological data RH (in blue) and temperature (in red) from RS launch at 10:58 UTC, and lidar data $\beta^{aer}$ from (13:40–15:10) UTC (in black). The resolution of the lidar data is 10 min and 7.5 m.

## 5. Conclusions

In our studies, we found higher backscatter values in spring 2020 compared to 2019, especially in the lower troposphere up to $\approx$1500 m in February. The cases with the highest backscatter showed very similar intensive optical parameters such as depolarization (below 2%), lidar ratio (around 29 sr at 355 nm) and Ångström exponent (around 1 for 355 nm and 532 nm), so we do not expect distinct aerosol properties, but rather "more from the same aerosol". The early *Arctic haze* season in February (in the lowest troposphere) was not accompanied by a higher lidar ratio. Instead, a clear case with a high lidar ratio may be explained by hygroscopic growth of aerosol rather than by a higher amount of continental aerosol. Hence, from the lidar ratio alone and without judging the meteorologic conditions,

the interpretation of lidar data may be misleading. We want to highlight this result because it may have implications for an appropriate definition of the *Arctic haze* in the future: either via high optical turbidity of dry aerosol regardless of origin, by hygroscopic growth of arbitrary particles or via a higher concentration of continental aerosol (the traditional *Arctic haze* consisting mainly of non-sea salt sulphates and anthropogenic markers). In this study, we did not find a clear indication of the latter type. Only an inversion of the aerosol microphysics using all lidar coefficients provides a realistic estimation of the origin and properties of aerosol.

The retrieval of the microphysical parameters confirmed these results and showed very distinct aerosol properties within the low troposphere on 13 January and 21 February. On 13 January, the most important aspects were a high lidar ratio of $LR^{aer}_{532\,nm} = 77.59\,sr$, a small refractive index of $RI_{real} = 1.311$ and a large effective radius of $r_{eff} = 0.73\,\mu m$. On this day, we probably had the same air masses as the Polarstern about 29 h ago, so a prospective comparison between both lidar measurements could be interesting. 21 February was the day with the highest backscatter coefficient at (700–1000) m. In the layer a bit higher (but still within the MBL) where we calculated the microphysical parameters, we found $RI_{real} = 1.526$ and $r_{eff} = 0.0799\,\mu m$ with a lidar ratio of $LR^{aer}_{532\,nm} = 39.33\,sr$. On both days, we also studied a layer above the MBL in the free troposphere. Here, the aerosol properties are very similar, with a refractive index of $RI_{real} \approx 1.45$ and a radius of $r_{eff} \approx 0.053\,\mu m$. The generally uniform optical parameters at this altitude indicate that the aerosol properties are not only similar for both case studies, but are stable for the whole period.

The mechanical boundary layer is characterized by a change in the backscatter coefficients. Wind shear, elevated temperature inversions and an increased relative humidity leads to an altered size distribution and refractive index. This point needs more attention in the future, since it is non-trivial to compare the results with regional climate models and/or ground based in situ measurements on a site such as Svalbard.

The main conclusions based on our results are:

- In 2020, aerosol backscatter below 1.5 km was found to be much higher than in 2019. Above that altitude, clear conditions with similar aerosol properties prevailed in both years. We found a dominance of small particles with radii below 100 nm. The almost constant aerosol properties above 2 km altitude suggest, if confirmed at other sites, that, in principle, regional climate models might be easily fed with realistic aerosol properties above this altitude for the Arctic;
- Even in the MOSAiC winter with additional meteorologic data, air backtrajectories alone may not be reliable (high and low aerosol for similar air masses from Siberia). Hence, a final proof of why 2020 was more turbid cannot be given;
- Backscatter histograms for 2020 and low altitudes show a bi-modal structure but the average LR and Ångström exponent for those high and low backscatter groups are very similar. Hence, high backscatter means usually "more of the same aerosol";
- We generally found low aerosol depolarization. The dominance of nearly spherical particles means that Mie theory is justified to connect optical and microphysical aerosol properties;
- We found low to moderate RI (from four case studies only);
- The highest LR was found for a case with high humidity and low refractive index: likely a case of hygroscopic growth. This means that the LR alone, without knowledge of humidity, is not a good indicator of aerosol type in the Arctic;
- Similarly, other cases of high LR were already found in January for days with lower than average backscatter;
- The low depolarization, the low to moderate RI and the possibly hydrophilic behavior is in agreement with ground-based in situ observations showing nss-sulphate and marine aerosol to be the dominant aerosol species in this season:
- There is generally much higher backscatter and more variable aerosol properties below 2 km in altitude. Bi-modal volume distribution functions can occur. We found clear indications that (at least part) of this aerosol variability in the lowest 2 km is

connected to elevated temperature inversions or gradients of humidity. This possible modification of aerosol properties over the undisturbed Arctic oceans compared to the local measurements over Svalbard needs more attention in the future.

**Author Contributions:** Evaluation of the lidar data was performed by J.D.; the inversion of the lidar data was performed by C.B.; the manuscript has been written by J.D.; C.R., as PI of the instruments, acted as supervisor. All authors have read and agreed to the published version of the manuscript.

**Funding:** This research received no external funding.

**Data Availability Statement:** The data of ths work is freely availably from the authors.

**Acknowledgments:** The lidar measurements were performed by Sandra Graßl, Fieke Rader and Wilfried Ruhe.

**Conflicts of Interest:** The authors declare no conflict of interest.

## Appendix A. Ångström Exponent

The Ångström exponent $Å$ (or the color ratio $CR$) is often used for a rough estimation of the particle radius $r$. It is expected that particles with a small size parameter $x = \frac{2\pi r}{\lambda}$ have an exponent of $Å \approx 4$ due to Rayleigh scattering, while large particles ($x \gg 1$) have an exponent of $Å \approx 0$. Hence, it is expected that the Ångström exponent increases with altitude due to decreasing particle size. However, using photometer data, it was shown that the assumption of one single Ångström exponent may be only a rough approximation for Arctic aerosol [10]. Hence, the exponent was calculated by Equation (1) for all pairs of wavelength $\lambda$. All three calculated Ångström exponents obtained different results and even the trend with the altitude differs. As can be seen in Figure A1, $Å$ increases with altitude when it is calculated with $\lambda_1 = 355\,\text{nm}$ and $\lambda_2 = 532\,\text{nm}$, while it decreases when calculated with $\lambda_1 = 532\,\text{nm}$ and $\lambda_2 = 1064\,\text{nm}$. It was checked whether the calibration of the 1064 nm channel could be the reason for the difference, but no significant change could be observed by changing the boundary conditions of this channel. As the wavelength difference is smaller between $\lambda_1 = 355\,\text{nm}$ and $\lambda_2 = 532\,\text{nm}$, this wavelength pair should be best suited for the determination of a constant Ångström exponent. Indeed, the exponent $Å_{(355,532)}$ fits best to the expectation of decreasing particle size with altitude. The same result was also found by [29]. Hence, we propose to always calculate the Ångström exponent from close wavelengths and avoid the infrared region. Nevertheless, the results show that the calculation of the Ångström exponent is critical, and interpretation as the particle size should be performed with caution [10].

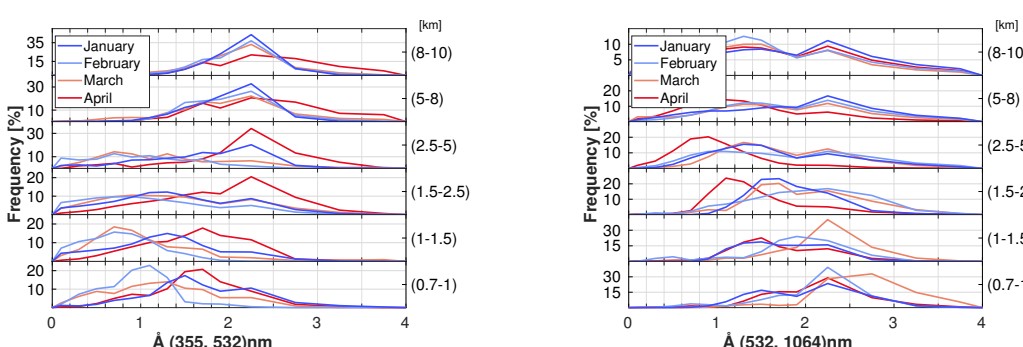

**Figure A1.** Frequencypolygon of the Ångström exponent for different height intervals and months. Left: calculated with 355 nm and 532 nm; right: calculated with 532 nm and 1064 nm. The trend differs for different wavelength pairs.

## Appendix B. Backward Trajectories

It is known that backward trajectories in the Arctic should be used with caution because of the low availability of meteorological data. Earlier studies showed that only a few additional meteorological measurements can improve the weather forecast in the Arctic

significantly [38]. Hence, it was expected that the additional observations from the MOSAiC expedition in the central Arctic and also the additional radiosonde launches in Ny-Ålesund during that time would also lead to an improvement of the backward trajectories. However, our analysis showed that there is not such a clear, immediate improvement for the MOSAiC winter. When calculating the trajectories with different models, clear differences occurred, as can be seen in Figure A2 on the left side. Especially on 21 February, there were clear differences in the origin of the air masses ranging from Greenland to Siberia. Such problems occurred primarily in the direct vicinity of cyclones. Trajectory clusters are a good option to proof the stability of the trajectories and to observe such problems. On the right side of Figure A2, the deviation of the *GDAS1* and the *Reanalysis1* models using HYSPLIT is plotted against time. Both models are commonly used in aerosol sciences. For Figure A2, trajectories were calculated every 12 h with both models for the whole months of January 2019 and January 2020, and the mean distance between the trajectories of both models were calculated. This was done for two different altitudes: one within the MBL and one in the free troposphere (with greater wind speeds). It can clearly be seen that the deviation between both models did not decrease for the MOSAiC year. Apparently, the additional data from MOSAiC in 2020 does not lead to a significant improvement compared to the previous year. After 2 days, there is a difference (proxy for insecurity) of about 500 km, and after 5 days, we have more than a 2000 km difference between both models, so air trajectories should always be used with great caution in the Arctic environment.

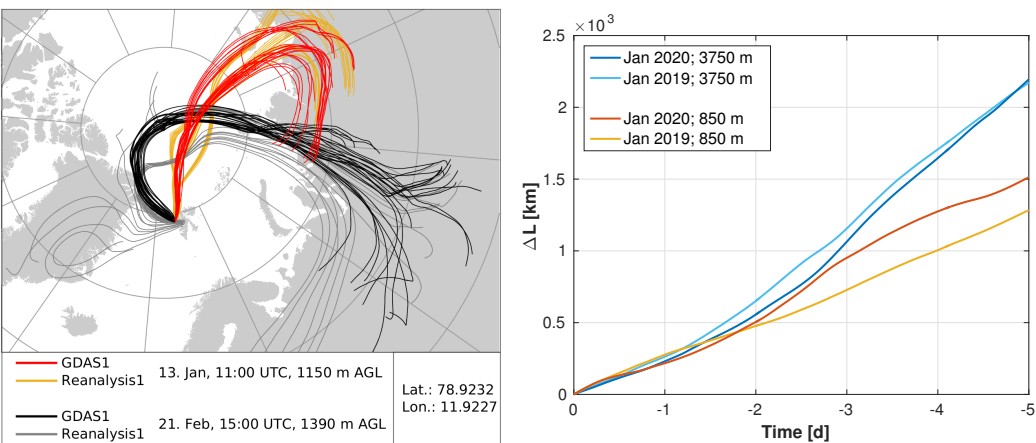

**Figure A2.** (**Left**): Comparison between the trajectories calculated with Reanalysis1 and with GDAS1. Both models used the same start parameters; (**Right**): mean difference between the trajectories of GDAS1 and Reanalysis1. All trajectories were calculated for the whole month every 12 h with the same start parameters.

## Appendix C. Error Propagation from Optical to Microphysical Properties

Here, we give a short, preliminary discussion of how errors in the lidar input variables may affect the inversion of the microphysical parameters. As an example, the lower layer of the case of 21 February has been considered (1323 m to 1586 m altitude, Table 3). In total, 10 additional inversions were performed, which are shown in Table A1. For this experiment, the lidar input data was modified by an error of 60% of the maximum error of Section 4.1. Recall that for the undisturbed solution (first line in Table A1), the lidar ratios for 355 and 532 nm were almost equal (around 39 sr). It can be seen from the table that the inversion results are quite consistent. In particular, the single scattering albedos for both colors are always close to 0.85. Further, for most cases, the bi-modal size distribution with the dominance of the small particle mode was reproduced nicely (not shown, but can be understood from the effective radius in Table A1). However, whenever the aerosol extinction at 355 nm is overestimated and at the same time the extinction at 532 nm is underestimated (lines in which the effective radius is given in bold in Table A1), the volume distribution function changes significantly into a broad one-modal distribution with a

maximum at radii $\geq 0.3$ µm. As we have no reason to assume that such a systematic error in the retrieval of the aerosol extinction could have occurred, we rate the existence of large particles as unlikely.

However, clearly such an error propagation from the optical into the microphysical properties must be analyzed in more detail in the future.

**Table A1.** Microphysical parameters for the low layer on 21 February 2020, considering errors in the lidar input data.

| Error Realization | Real (RI) | Imag (RI) | $r_{eff}$ | $v_t$ | $SSA_{355}$ | $SSA_{532}$ |
|---|---|---|---|---|---|---|
| Exact solution | 1.526 | 0.020 | 0.0799 | 6.16 | 0.8540 | 0.8620 |
| $\alpha^{aer}_{355\,nm}$ high, $\alpha^{aer}_{532\,nm}$ high | 1.520 | 0.021 | 0.0720 | 7.18 | 0.8553 | 0.8541 |
| $\alpha^{aer}_{355\,nm}$ low, $\alpha^{aer}_{532\,nm}$ low | 1.538 | 0.020 | 0.1010 | 5.25 | 0.8514 | 0.8703 |
| $\alpha^{aer}_{355\,nm}$ high, $\alpha^{aer}_{532\,nm}$ low | 1.533 | 0.020 | **0.3529** | 6.27 | 0.84128 | 0.87294 |
| $\alpha^{aer}_{355\,nm}$ low, $\alpha^{aer}_{532\,nm}$ high | 1.561 | 0.022 | 0.0511 | 6.66 | 0.86155 | 0.8389 |
| All $\beta^{aer}$ low | 1.517 | 0.020 | 0.0865 | 6.30 | 0.8529 | 0.8609 |
| All $\beta^{aer}$ high | 1.529 | 0.020 | 0.0865 | 6.16 | 0.8508 | 0.8585 |
| All $\beta^{aer}$ low, $\alpha^{aer}_{355\,nm}$ high, $\alpha^{aer}_{532\,nm}$ low | 1.520 | 0.017 | **0.3991** | 7.14 | 0.8515 | 0.8800 |
| All $\beta^{aer}$ low, $\alpha^{aer}_{355\,nm}$ low, $\alpha^{aer}_{532\,nm}$ high | 1.547 | 0.021 | 0.0551 | 7.34 | 0.8585 | 0.8347 |
| All $\beta^{aer}$ high, $\alpha^{aer}_{355\,nm}$ high, $\alpha^{aer}_{532\,nm}$ low | 1.537 | 0.021 | **0.2959** | 5.64 | 0.8337 | 0.8678 |
| All $\beta^{aer}$ high, $\alpha^{aer}_{355\,nm}$ low, $\alpha^{aer}_{532\,nm}$ high | 1.581 | 0.022 | 0.0505 | 6.35 | 0.8684 | 0.8501 |
| Mean value | 1.537 | 0.02 | 0.1482 | 6.40 | 0.8528 | 0.8591 |
| Standard deviation | 0.02 | 0.001 | 0.1321 | 0.64 | 0.0093 | 0.014 |

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
