# Peer review of "Lidar-Derived Aerosol Properties from Ny-Ålesund, Svalbard during the MOSAiC Spring 2020"

_remotesensing, doi:10.3390/rs14112578_

Round 1

Reviewer 1 Report

In my opinion the revised version of the manuscript "Lidar derived Aerosol properties from Ny-Alesund, Svalbard during the MOSAiC spring 2020" by J. Dube, C. Bochmann and C. Ritter suffers from the same main problems characterizing the first submitted version. Even if some of the improvements I suggested have been considered by the authors some others (listed below and in my opinion the most important ones!) have been not addressed from practical point of view. 

Lack of estimation of statistical uncertainties on all presented aerosol optical properties

Concerning this point the authors state that statistical uncertainties are not reported because the tool they have been used to process lidar measurements does support such calculation. Frankly I don't think this can be considered a valid reason to exclude such an important aspect of the data analysis. In my opinion this is valid in general and not only for this work. So I strongly recommend the authors to implement the calculation of statistical uncertainties in the tool they use to process their lidar data. As I have already mentioned in my previous review, for any measurements (direct or indirect), it's fundamental to provide the corresponding error bars! Especially in this case where the procedures to implement are well documented in literature.

Despite this lack in their software analysis,  the authors have included in the table 3 a sort of error bars on optical properties saying that these represent the largest possible errors without providing any additional information (how these errors have been estimated? which statistics has been assumed?).  Moreover, they also state that this error estimation (whatever it comes from) has been not included in the calculation of uncertainties on microphysical properties (which consequently are underestimated). Frankly, I see this procedure very confusing and also not fully correct from methodological point of view.

Results not presented properly

In my previous review, I suggested to provide more information (which in principle should be available to the authors according to the described lidar specs) beside the backscatter at 532, the LR at 355 and the depol at 532 when they compare the observations during 2020 against the ones during 2019.  In this way I would be possible to discuss in more complete and detailed way the differences or the similarities with respect to 2019. The corresponding part of the manuscript was not modified at all by the authors and moreover they did not provide any motivation for which only some of the properties they measured have been taken into account. 

Moreover, I suggested to provide the 3+2+2 lidar analysis (in terms of vertical profiles) from which the optical properties reported in table 3 are derived.  The authors added two figures (5 and 6) in which they show lidar profiles at different wavelengths which is something not very useful (basically looking at these figures is not possible to draw any relevant conclusion concerning the atmospheric conditions occurring during the selected cases) and very different from what I suggested to add. As 3+2+2 lidar analysis I meant a set of plots in which the integrated (in the same time windows the quantities in table 3 refer to) 3 backscatters, 2 extinctions and 2 particle/volume depolarization (and all the corresponding intensive properties like LRs and angstrom exponents) are reported as vertical profiles (possibly at altitude a bit higher than 3.5km).  

According to the above substantial problems, in my opinion the revised manuscript is still far to be in a form suitable for acceptance.

Reviewer 2 Report

Review of the revised manuscript "Lidar derived Aerosol properties from Ny-AÌŠlesund, Svalbard during the MOSAiC spring 2020" by Jonas Dube et al. submitted to Remote Sensing

The manuscript presents the results of a Raman lidar sounding campaign in Ny-AÌŠlesund, Svalbard, using available lidar equipment and already published analysis techniques. The campaign aims at characterizing aerosol properties as these are, according to the authors, not very well understood in Svalbard, and may help with the definition of Arctic Haze.

I strongly support publication of results and conclusions of all remote sensing experiments, that have scientific merit in Remote Sensing, and this work definitely has the potential for such a publication. Unfortunately, while the paper presents the results of atmospheric characterization, it fails to make any significant conclusions (or point out specific instrumentation / analysis advances), which would be the true added value to science of this work. I would strongly recommend the authors to re-think what are specific advances to the understanding of the atmospheric conditions and point them out clearly. I recommend a major revision of the manuscript.

Specific comments

  • In section 2 the authors describe the lidar they used and the methods they applied to extract atmospheric properties. Here, it is not clear whether they used standard (already published) methods only or they used them as a basis for their own. This should be more transparent.

  • While details of experimental setup are given in reference [18], it would still be worthwhile to state the operating conditions of the device, in particular the settings of the Licel transient recorders, the firing rate of the laser, etc.

  • From Jan 9 – April 26 2020 campaign, (107 days) only 2954 time steps (23.6 days) of data were taken, which is roughly 1/5. Why? Is this because of adverse weather conditions in 4/5 of the time? Was the 2954 time steps data further reduced due to “cloud mask”, etc? More details on data handling should be given.

  • In section 4 (case studies), the authors go into details about the retrieved values for specific cases, but these details can not really explain the atmospheric processes at hand at that time (and give rise to speculations, e.g. lines 200-223). I would suggest a more concise reporting on the results and more concrete conclusions in the final section, separated from results.

  • The main conclusions in section 5 are:

    • Higher backscatter in 2020 than in 2019, offering no clear explanation as to why. The authors guess at hygroscopic growth and at the same time state the absence of “continental” (dust?) aerosols. Which type of aerosols experienced the growth then? Maritime? Why?

    • Decrease of the mean of particle size distribution with altitude is not unexpected, it was observed and documented under many different conditions (even in the investigation of volcanic plume). Please clarify why is such an observation meaningful in your case.

  • The authors are not consistent with citations, as they sometimes cite papers in brackets only (e.g. [33] on line 230), and at other times by adding the name of the first author (e.g. lines 35-36). This is in my opinion not needed, the details on the authors are available in the references. Above all, I advise the authors to be consistent and uniform with citations.

Reviewer 3 Report

This is an excellent paper presenting detailed and extensive Raman lidar backscatter measurements of the atmosphere.   The results are important and novel to the atmospheric community.    The authors do a good job in detailing the backscatter parameters, field data, and calculated optical and concentration values.   My only suggested changes are to add to the general usefulness of the paper and include these optional changes:  (1) Under Instruments....sentence with "1064 nm with 50MHz vertically " does not make sense....is this the bandwidth or linewidth; (2) Might give some of the basic laser/optics parameters in the paper (as opposed to "see Ref. 18"); (3) detail the laser wavelengths, Raman emission wavelengths and from what species; (4)  maybe Indicate more parameter values and quantified findings in the Abstract (maybe 3 or 4 values, such as those in the Conclusions)....also say in the first sentence of the abstract....three wavelength Raman Lidar (.355, .532, 1.06 microns).

Round 2

Reviewer 1 Report

Looking at some answers the authors provided I realized that probably some misunderstanding occurred during the previous rounds. I will try to provide some more clarifications here.

Uncertainties estimation

1) Of course I cannot know the details of the analysis tool the authors used to calculate aerosol optical properties. My comments were just triggered by some sentences (in my opinion not fully clear!) the authors wrote.

For example if the authors state "Since it is not possible to include the errors of the optical measurements in the retrieval software, the uncertainties of the microphysical parameters only contain the uncertainty from the inversion method and are therefore underestimated", a reader (and this was my case actually) could  perhaps understand that the retrieval software mentioned here is the one used to compute the aerosol optical properties. Now, looking at the authors' answers, I realized that probably they mean the microphysical retrieval one. Anyway, in my opinion, without this additional explanation it is not possible to fully understand what the authors wanted to mean exactly. In the current version of the manuscript, this sentence has been modified in: "Since it is hardly possible to include the errors of the optical measurements in the retrieval software, the uncertainties of the microphysical parameters only contain the uncertainty from the inversion method and are therefore underestimated.", which I still think can be misunderstood. So I would suggest to clearly state somewhere in the manuscript what the authors mean with "retrieval software". 

2) Another sentence I found not clear is:  "For the optical parameters the largest possible error according to error propagation is given". According to my knowledge, the error propagation provides a method to estimate THE uncertainty/error not the "largest possible error". Usually expressions like "largest possible error" are used to indicate different concepts. Or maybe as "largest possible error" the authors want to mean the 3-sigma interval? If this is the case why the authors did not just provide the more common standard deviation? Is this because they are attempting to combine systematic and statistical uncertainties? I think it is essential:

a) to clarify what the authors intend as "largest possible error" exactly

b) if this is something different from the standard deviation they should explain why they have used it and how it has been calculated (maybe providing specific references)

c) maybe think to a better (more standard) nomenclature if possible.  

Figure 5 and 6

These "new" figures still provide only a partial view of the atmospheric aerosol contents as the intensive parameters are missing. I report here once more what I have already suggested in the previous round: "As 3+2+2 lidar analysis I meant a set of plots in which the integrated (in the same time windows the quantities in table 3 refer to) 3 backscatters, 2 extinctions and 2
particle/volume depolarization (and all the corresponding intensive properties like LRs and angstrom exponents) are reported as vertical profiles (possibly at altitude a bit higher than 3.5km).". Would be possible for the authors to just add this? As suggested, if the SNR is not high enough they could average in time the single profiles reported in the current figures. 

Appendix C

1) Even if this is only a first basic approach to the problem, it goes into the direction I was trying to suggest and I'm happy that the authors decided to add it. 

2) Why the authors decided to considered only 60% of the estimated error. Please clarify.  

3)  "As we have no reason to assume, that such a systematic error in the retrieval of the aerosol extinction could have occurred we rate the existence of large particles as unlikely".  This is again something not clear and by the way a different term "systematic error" has been used to indicate a quantity previously indicated in a different way! Moreover, if it is true that the authors have no reasons to assume the errors are as large as they have estimated, it means that their estimation is most probably wrong... And this has nothing to do with the complexity to propagate uncertainties from optical to microphysical quantities. This is just a problem of error estimation on the optical quantities. If, on the other hand, the authors assume their error estimation on optical quantities is robust than they should simply accept that there is a probability (which actually they are not able to quantify!) that large particles could be really present into the atmosphere. All the rest is more or less arbitrary. I suggest to re-phrase accordingly.     

Author Response

Please find our detailes answer attached

Reviewer 2 Report

Review of the second revision of the manuscript "Lidar derived Aerosol properties from Ny-AÌŠlesund, Svalbard during the MOSAiC spring 2020" by Jonas Dube et al. submitted to Remote Sensing

I find that the authors have considerably improved the manuscript, including its conclusions, and recommend its publication after a minor revision addressing a few specific comments pointed out below.

Remaining inconsistencies in citing

Throughout the manuscript there are citations without mentioning the authors names, except for:

  • Line 36-37: However, Warneke et al. [16] showed by aircraft measurements over Alaska that biomass burning also contributes to Arctic Haze. → However, it has been shown by aircraft measurements over Alaska that biomass burning also contributes to Arctic Haze [16].

  • Line 113-114: For a better comparison also the values for 2019 are shown in grey, which were evaluated by Rader et al. [29]. → For a better comparison also the values for 2019 are shown in grey, which were evaluated by Rader et al. [29]. → For a better comparison previously published values for January-March 2019 are also shown [29].

    Details about the plotting (e.g. color) should be in the caption.

  • Line 118-119: This is effected by a higher aerosol concentration in the stratosphere due to wildfire smoke, which is already described by Engelmann et al. [6]. → This is effected by a higher aerosol concentration in the stratosphere due to wildfire smoke [6].

  • Line 259-261: Zieger et al. [37] found, based on in-situ measurements, that in Ny-AÌŠlesund already relative humidities around 50 % are sufficient for hygroscopic growth of the aerosol. → Based on in-situ measurements it was found that in Ny-AÌŠlesund relative humidities around 50 % are already sufficient for hygroscopic aerosol growth [37].

Figure and table captions

  • Figure 1-3: I recommend that all the text from the first paragraph of Section 3 that describes the figures (e.g. colors of lines, etc.) is moved into figure captions.

  • Table 1. Median of the lidar ratio LR for different altitude ranges and month separated for all high and small backscatter events. In the brackets also the 25. and 75. percentile are indicated. → Table 1. Median of the lidar ratio (LR) for different altitude ranges and months in 2020 separately presented for all high and small backscatter events. The 25. and 75. percentile are indicated in the brackets.

  • Table 2. Median and in brackets the 25. and 75. percentile of different optical parameters for the 3 days in February with the highest backscatter and the rest of February. → Table 2. Median of the retrieved optical parameters shown separately for the 3 days in February 2020 with the highest backscatter and the rest of February. The 25. and 75. percentile are indicated in the brackets.

  • Figures 5 and 6: I recommend that the text from the first paragraph of Section 4.1 that describes the details of the figures (e.g. colors of lines, etc.) is moved into figure captions. As it is, the details break the red line of the manuscript, they are only needed when looking at the plots.

Other

  • Lines 160-162: Please rephrase the starting paragraph of Section 4, perhaps like:

    As already mentioned above events with a high backscatter coefficient and with a high lidar ratio have to be distinguished. To get more information about the similarities and differences between the aerosols at both occurrences we investigated two case studies. → To get more information about the similarities and differences between the aerosols in the events with a high backscatter coefficient and with a high lidar ratio two representative cases are presented in detail.

  • Arctic Haze is written in a haphazard way throughout the paper, including: Arctic Haze, Arctic Haze, «Arctic Haze» - it may be a good idea to decide on one

Conclusions

  • I am still not fully happy with the conclusions section, which is a mixture of the overview of the research performed, reporting of important results (values) and the conclusions. I would recommend moving the results to the previous section, minimizing the review and focusing on the conclusions. The bullet list is fine, but please make it textually consistent (some bullets start with a capital letter, some do not, some bullets end with a punctuation mark, some do not…) As an example, I recommend the following change:

    Finally, we summarize our results like follows. We found:

    • in 2020 higher backscatter than 2019 below 1.5 km and above similar and clear conditions. We found a dominance of small particles with radii clearly below 100 nm. The almost constant aerosol properties above 2 km altitude mean, if confirmed over other sites, that in principle regional climate models might be easily fed with realistic aerosol properties above this altitude for the Arctic →

    The main conclusions based on our results are:

    • In 2020 aerosol backscatter below 1.5 km in similar atmospheric conditions was found to be much higher than in 2019. We found a dominance of small particles with radii below 100 nm. The almost unchanging aerosol properties above 2 km of altitude suggest, if confirmed at other sites, that in principle regional climate models might be easily fed with realistic aerosol properties above this altitude for the Arctic.

    • .

Author Response

PLease find our answers attached

Round 3

Reviewer 1 Report

I would like to thank the authors for their responses and for addressing most of my comments in their revised manuscript.  I believe the paper has been significantly improved and thus it can be accepted for publication.

This manuscript is a resubmission of an earlier submission. The following is a list of the peer review reports and author responses from that submission.

Round 1

Reviewer 1 Report

This paper describes about research of the Arctic Haze (AH) based on lidar measurements over the spring months in 2020. Two events in Jan 13 and Feb 21 are discussed involving parameters of backscattering and lidar ratio with very confusing writings. For example, initially they give six height ranges then in Section 4 discussion is made in four different heights, finally there is boundary layer height.  Maybe the main problem is the limited data they obtained making the identification of AH difficult.  Other problems seem to be their regarding AH as a kind of pollution with a well defined parameter and also for the continental aerosol.

According to what I found in literatures, AH, or continental aerosols, corresponds to extinction and low visibility conditions in the Arctic due to pollution transported from Eastern Europe including dust, industrial and traffic pollution, biomass burnings, and ocean aerosols etc. each with different lidar ratios and other optical properties.   From reference 25 cited in the paper, the lidar ratio of AH may be in the range of 35-70 covering a many different types of aerosols. In Table 4, LR’s are ranged 77-39 for two events in Jan and Feb which are suitable for AH caused by various kinds of aerosols. For high LR, this maybe related with biomass burning aerosols.

 There are many uncertainties in addition to above problems, the Backtrajectories show aerosols of Jan 13 traced to coastal area which may be dominated by maritime aerosols  of low LR with coarse particles instead of high LR as reported here, whereas the high reff reported here is consistent sea salt aerosols. For the Feb 21 case, backtrajectories traced to inland of continental aerosols of various kinds aerosols with LR of 30-40 reasonable.  In the last section, the discussion of high humidity conditions further complicates the problems without definite answers.

A consistent result for this paper may be the small depolarization ratios for all cases.  This indicates small particles (not spherical particles) due to pollution.

Another major problem is lack of background information about AH and the literature survey are very brief leaving the readers to find out by themselves. With these problems, I think a lot of works needed to be done to be accepted. The leading author need to read more literature understanding various types of aerosols.

Reviewer 2 Report

General comments:

Manuscript presents the results of multiwavelength Raman lidar sounding of the Artic Haze events during time January to April 2020.

Influence of large scale aerosol transport to Arctic regions on snow pollution and melting process is an understudied process. In this study, the authors carry out a detailed investigation of the optical and microstructural characteristics of the aerosol in Svalbard using data from the Ny-Alesund Raman lidar station. The topic of the manuscript and its content is appropriate for publication in Remote Sensing.

The choice of the procedure and data processing algorithms for retrieving a set of optical parameters and particle volume size distributions (PVSD) is fully justified.

The description and interpretation of the results are sufficiently detailed. The complex process of PVSD transformation is explained quite logically.

Specific comments:

  1. Figure 1 shows the grid of the imaginary and real part of the refractive index. All the same, how did you choose the optimal values of a pair of parameters (CRM, CRI) presented in table 4 ?

  1. The manuscript deals with lidar data for the period from January to April 2020. The AERONET radiometric station Ny_Alesund_AWI started measurements in March, 2020. Since April, it presents the results of retrieving a full set of aerosol optical parameters and PVSD, averaged over the aerosol layer. Have you checked the consistency of the sets of aerosol parameters obtained from the data of lidar and radiometric measurements, taking into account the specifics of the measurement methods?

Reviewer 3 Report

The manuscript "Lidar derived Aerosol properties from Ny-Alesund, Svalbard during the MOSAiC spring 2020" by J. Dube, C. Bochmann and C. Ritter provides a climatological analysis of the observations made by KARL, the lidar system operating at Rench-German AWIPE research station located in Svalbard. In the first part of the manuscript the authors compare, at different altitude ranges, the aerosol properties they have retrieved in the period January - April 2020 (called spring 2020) against the ones obtained in the same period in 2019. They conclude that major differences are observed in the lower atmosphere as expected. Artic Haze events were identified as the ones corresponding to the highest value of particle backscatter coefficient and characterized in terms of both optical and microphysical properties. According to this analysis, the authors state that Artic Haze events are not characterized by an increase of LR and consequently they don't have to be linked necessarily to pollution events from continental aerosol. Moreover, on the base of intensive optical and microphysical properties and on backtrajectories, the authors infer that the cases characterized by high values of LR are due to hygroscopic process of arbitrary particles independently from they origin. Even if the topic is quite interesting especially considering the location the observations, in my opinion the manuscript suffers from several critic problems. So I strongly encourage the authors to resubmit it after having addressed the points listed below.

Major points:

  1. The authors state (lines 73-75) "Since in almost all cases the noise level of the backscatter and extinction profile is often not known or only a roughly estimation is available, the parameter choice rule consists in selected an appropriate trible (n,d,k) heuristically". I totally disagree on that! Actually any measurement is not meaningful without an appropriate estimation of the corresponding uncertainties and lidar observation cannot be an exception to that fundamental rule. Indeed it's a good practice to include always in the lidar retrievals (for any optical property) the estimation at least of statistical uncertainties. In literature there are many examples on how this can be done following standard uncertainties propagation rules of applying more complex approaches like for example Monte Carlo simulations. Systematic uncertainties are a bit more complex to handle but usually they are minimized correcting the raw lidar signals for instrumental effects. So I would suggest the authors to include the estimation of statistical uncertainties in their lidar retrieval and use them to constrain the microphysical retrieval instead of doing it more or less arbitrarily. 
  2. The same concern on the evaluation of statistical uncertainties applies also to the data the authors present in table 4. All the optical properties are reported without error bars which in my opinion is not acceptable. Once again any comparison of these values without an indication of the corresponding error bars is not meaningful! Moreover, it is quite strange to provide error bars on quantities like refractive index, volume concentration,... and not for the optical parameters used to derive them... Obviously such error bars do not include all the sources of uncertainties and so, most probably, they are underestimated. If it is not possible to include in the microphysical retrieval the statistical uncertainties on the optical parameters the authors should at least mention that what they provide as error bars on microphysical parameters is an underestimation (and maybe they should provide more details on how they calculate them).
  3. In general, I found the way in which the results are presented a bit confusing. The authors states that the lidar measurements they have considered for this study include 3 elastic and 3 Raman wavelengths and 2 depolarizations. However, in the the first part of the study apparently only the backscatter at 532 and the LR at 355 are used to compare the observations during 2020 against the ones during 2019. Why only these two properties has been selected among all the other available? The authors do not provide any explanation about that. I would suggest to include in the comparison all the properties they have available and discuss more in general the differences or the similarities with respect to 2019. If this is not possible the authors should provide a valid motivation. 
  4. In the second part of the manuscript two cases study are presented. However, the authors do not provide to the reader an easy way to "visualize" these cases study. They just provide the values in table 4 and some plots mainly related to the microphysical properties and to the estimation of MBL. I think it is necessary to provide the 3+2+2 lidar analysis (in terms of vertical profiles) from which the optical properties reported in table 4 are derived. This would give a much clearer picture of the atmospheric conditions including the position of the layers and the stability of the optical properties within these layers.

Less relevant points:

  1. Some figures or tables not fully described in the text of in the corresponding captions.  For example in figure 7 and 8 are reported several backscatter profiles. At which times each of one refer to? What is the integration time used for to retrieve them? Moreover, they clearly show some regular ringing... From where this distortion come from? In the same way the table 4 shows some value for optical parameters. What exactly these values represent? Mean value withing the layer? This should be clearly specified.      
  2. The authors identified several atmospherical altitude ranges in which evaluate optical properties (H1-H6) without providing any information about the criteria used to identify such ranges. Are these altitude ranges selected taking into account the positions of the layers in the considered period? Is the climatology of MPL of the measurement site somehow taken into account? Another interesting possibility would be to compare the position of the layers occurred in 2019 and 2020 (especially for that it concerns the MBL height).
  3. The authors states that they apply a cloud mask to the lidar measurements to avoid bias in aerosol statistics. However, they do not provide enough details on how the mask is applied. I think it is important to better specify this point first of all because of the very particular location and moreover it should be somehow proved  that low values of LR sometimes reported in the manuscript cannot be due to residual cloud contamination.
  4. No details are provided concerning the estimation of the MBL heights. Which methods the authors have used to derive it? Do they look at the edge of particle backscatter or at the temperature inversion? Else? 
  5. The authors infer a possible hygroscopic growth occurring within the MBL. If this is true it would be useful to see a plot of the LR versus RH (both within the MBL) for all the period considered to check for possible correlations. 
  6. Strictly speaking the Angstrom exponent is defined considering the particle extinction (or the optical thickness) at different wavelengths. The corresponding one calculated using the particle backscatter is usually called "Backscatter-related Angstrom exponent" or "Backscatter Angstrom exponent". I would suggest to use the same nomenclature. 
  7. In table 2 and 3 the median is reported. Why the authors have decided to use this statistical parameter and not for example the mean? I can imagine several reasons but the authors should better motive their choice.
  8. Could the difference in the calculation of A(355,532) and A(532,1064) be somehow due to inaccuracy in the calibration of 1064nm. It is well know that the calibration of this channel is quite tricky. As the authors added an appendix just to describe that with a quite important final statement,  I would suggest to add there also the motivations for which they can exclude that, in their case, this is not an effect of a not fully accurate calibration of the infrared channel.